# Neural Correlates of Impaired Cognitive Control in Individuals with Methamphetamine Dependence: An fMRI Study

**DOI:** 10.3390/brainsci13020197

**Published:** 2023-01-24

**Authors:** Ani Zerekidze, Meng Li, Nooshin Javaheripour, Laura Huff, Thomas Weiss, Martin Walter, Gerd Wagner

**Affiliations:** 1Department of Psychiatry and Psychotherapy, Jena University Hospital, 07743 Jena, Germany; 2Department of Clinical Psychology, Friedrich Schiller University, 07743 Jena, Germany

**Keywords:** crystal meth, methamphetamine dependence disorder, cognitive control, Stroop task, prefrontal cortex, anterior cingulate cortex, dorsal striatum, visual cortex, functional magnetic resonance imaging, fMRI

## Abstract

Impaired cognitive and behavioral control has often been observed in people who use methamphetamine (MA). However, a comprehensive understanding of the neural substrates underlying these impairments is still lacking. The goal of the present study was to study the neural correlates of impaired cognitive control in individuals with MA dependence according to DSM-IV criteria. Eighteen individuals with MA dependence and 21 healthy controls were investigated using Stroop task, fMRI, and an impulsivity questionnaire. Overall, patients were found to have significantly poorer accuracy on the Stroop task and higher self-rated impulsivity. Comparing brain activations during the task, decreased activation in the dorsolateral prefrontal cortex (DLPFC), anterior midcingulate cortex (aMCC), and dorsal striatum was observed in individuals with MA dependence, compared to healthy controls. Altered fMRI signal in DLPFC and aMCC significantly correlated with impaired behavioral task performance in individuals with MA dependence. Furthermore, significantly lower and pronounced brain activations in the MA group were additionally detected in several sensory cortical regions, i.e., in the visual, auditory, and somatosensory cortices. The results of the current study provide evidence for the negative impact of chronic crystal meth consumption on the proper functioning of the fronto-cingulate and striatal brain regions, presumably underlying the often-observed deficits in executive functions in individuals with MA use disorder. As a new finding, we also revealed abnormal activation in several sensory brain regions, suggesting the negative effect of MA use on the proper neural activity of these regions. This blunted activation could be the cause of the observed deficits in executive functions and the associated altered brain activation in higher-level brain networks.

## 1. Introduction

The abuse of methamphetamine (crystal meth) has markedly increased in the last decades [1], which gave rise to the discussion about its harmful effects on the brain function because of its strong and direct impact on the central nervous system (CNS); [2,3]. Methamphetamine (MA) is a highly addictive stimulant drug, with acute effects that are associated with a heightened sense of alertness, increased energy, suppression of appetite, decreased fatigue, as well as altered attention and concentration [4]. Previous studies showed that MA consumption leads to a marked increase in the levels of monoaminergic neurotransmitters: the dopamine (DA), noradrenaline (NA), and serotonergic (5-HT) in the CNS, as well as in the peripheral nervous system by multiple complex pharmacological mechanisms [5,6]. It inhibits monoamine reuptake transporters, as well as reverses transport of the neurotransmitter through plasma membrane transporters [7]. MA also inhibits the monoamine oxidase activity and increases the activity and expression of the tyrosine hydroxylase, which catalyzes the conversion of the amino acid L-tyrosine into levodopa, a DA precursor [8]. These mechanisms lead in total to a significant release of monoamines.

Regarding the DA system, MA activates the mesolimbic, mesocortical circuit, and the nigrostriatal pathways [5,9,10]. Drug-induced euphoria is assumed to occur primarily as a result of increased firing of dopamine neurons that causes a dopamine release to supraphysiological concentrations [11,12] and D2 receptor stimulation in the mesolimbic pathway [13,14].

Furthermore, the high density of noradrenergic (NA) fibers from NA-synthesizing neurons in locus coeruleus (LC) to the anterior cingulate cortex (ACC), the prefrontal cortex (PFC), and the hippocampus have been shown [15,16,17,18]. Thus, a massive release of NA after MA consumption would strongly affect the excitability of these regions, having an effect on arousal, memory, attention, and cognitive control processes [15,16,17,18].

Because of these wide-ranging neurochemical effects of MA intake, chronic abuse of MA has been related to alteration in several cognitive domains [5,19,20,21,22,23,24]. A meta-analysis of 18 studies summarized that individuals with methamphetamine use disorders showed medium size deficits in cognitive control functions involving response inhibition and problem-solving, but also in episodic memory and psychomotor functions [25]. The observation of abnormal cognitive control functions coincides well with the clinical observations, that individuals with MA use disorder tend to act impulsively and have difficulties in the inhibition of impulses [21,26,27]. Impulsivity is a multidimensional trait often described as a “predisposition for rapid, but often premature, actions without appropriate foresight” [28]. Its operationalization, therefore, comprises the inability to stop or withhold an ongoing or prepotent response or thought, despite anticipating adverse consequences, and the preference of a marginal immediate reward over a more significant but delayed one (difficulties in delayed gratification) [29]. For instance, in the delay discounting tasks, which assess the degree of self-control and impulsivity in decision-making, MA abusers often exhibited greater delay discounting and altered recruitment of frontoparietal regions, compared to controls [30].

Furthermore, cognitive inhibition, as an essential subcomponent of cognitive control, describes the process of suppression of the prepotent mental representations, involving unwanted thoughts, memories, perceptions, or emotions [31] to pursue the overarching goal. One of the common and highly validated psychological measures of cognitive control is the Stroop task assessing the ability to inhibit the prepotent answer, in which longer reaction times and higher error rates refers to the deficient inhibition of task-irrelevant answers. In the Stroop task, longer reaction times in the single Stroop task conditions and larger Stroop effect were reported in MA-dependent individuals, indicating abnormal cognitive inhibition processes [20,32,33,34].

However, other studies did not find any significant group differences regarding the Stroop interference in active users, compared to healthy controls, as well as to abstinent subjects [35,36]. Variability in the clinical status may partly explain these inconsistent results. For example, the clinical status of the subjects varied from study to study, including individuals in early abstinent [33], late abstinent [20], active users [36], or adolescent MA abusers [32].

Moreover, the brain regions crucially involved in higher-order control over behavior have been traditionally assumed to be the dorsolateral prefrontal cortex (DLPFC), the ventrolateral PFC (VLPFC), and the ACC. The functional role of the ACC is hypothesized to be the evaluation of actions and performance and indicating the need for behavioral adaptation and action revaluation [37]. At the same time, the DLPFC maintains the representation of the means to achieve the goal [38]. Both the NA and the DA systems are crucially involved in those cognitive control processes [16,39]. In our previous studies, impaired fronto-cingulate brain activation using the Stroop task was demonstrated in individuals with different mental disorders associated with abnormal monoamine neurotransmission [40,41,42].

Few previous functional neuroimaging studies investigated the neural foundation of impaired cognitive control processes in people with chronic MA consumption. Nestor and Ghahremani [33] showed that 10 early abstinent MA abusers exhibited less activation, compared to 18 healthy controls in PFC, ACC, and SMA only during the incongruent condition of the Stroop task using a block-design. In contrast, Salo and Ursu [43] reported a reduced conflict-related activation during incongruent Stroop trials in the right PFC and supplementary motor area (SMA) in 12 MA-dependent subjects, but no significant difference in the ACC was found, compared to 16 healthy controls. In the subsequent study, Salo and Fassbender [44] showed in a larger sample of MA abusers a negative correlation between the RT adjustment and PFC activation, but no differences in the ACC, as well as no significant correlations between the PFC activation and drug use pattern. Moreover, another study showed increased activation in the DLPFC, VLPFC, and inferior parietal lobule (IPL) in active MA-dependent individuals during the incongruent condition of the Stroop task, compared to the healthy controls, whereas no differences were found in the Stroop task performance [36]. In addition to the differences in clinical status, there were also differences in the study design and statistical analyses between studies, which may explain these conflicting results. For instance, some studies presented the stimuli using a short and fixed inter-trial interval (ITI) of 2500 msec [44], while Nestor and Ghahremani [33] and Jan and Lin [36] adopted a block design with congruent, incongruent, and rest blocks.

In the present study, the well-validated version of the Stroop task was used, which allowed us to detect robust BOLD activations in the fronto-cingulo-striatal regions [40,45]. Our goal was to investigate the neural correlates of impaired cognitive control in early abstinent subjects with MA dependence, administered to the inpatient ward for withdrawal therapy. We hypothesized that the MA group would show significantly more errors and longer reaction times in the Stroop task, in the incongruent, as well as in the interference conditions, compared to the control group. Furthermore, as the Stroop task performance was consistently associated with activation in DLPFC, ACC, and striatal regions, we expected to find a reduced BOLD signal in these brain regions in individuals with MA dependence, particularly in the incongruent condition. We also expected to find an association between altered brain activation and impaired behavioral performance in the Stroop task. In addition, based on the clinical observation of higher impulsivity, we hypothesized to find significantly higher scores in the self-reported impulsivity using the UPPS impulsive behavior scale [46].

## 2. Materials and Methods

### 2.1. Participants

A sample of 21 individuals with MA dependence and 21 healthy controls was recruited for the present study. As inclusion criteria, individuals with MA dependence must be older than 18 years and had to fulfill the criteria of the methamphetamine dependence, according to the DSM-IV criteria, which was established using the M.I.N.I. interview [47] by trained study personnel (LH). The exclusion criteria were comorbid schizophrenia and other psychotic, as well as affective disorders, neurological diseases of the central nervous system, and traumatic brain injury. Furthermore, patients, who fulfilled the criteria of other substance dependence in the last 12 months, were excluded from the study.

Healthy subjects who met criteria for substance abuse or dependence or other psychiatric disorders according to DSM-IV criteria as determined by the M.I.N.I. interview or who were diagnosed with neurological disorders assessed by a checklist were excluded from the study. Impulsivity was assessed by a German version of the impulsive behavior scale, UPPS [46], exploring four dimensions of impulsivity: lack of premeditation, urgency, sensation seeking, and lack of perseverance.

Three MA-dependent individuals could not complete the study due to non-adherence to the study protocol. The final sample consisted of 39 participants: 18 MA-dependent individuals (M_age_ = 32.4, SD = 7.4, range: 18–46 years) and 21 healthy controls (M_age_ = 27.6, SD = 3.5, range: 23–35 years). The MA group included 14 men (78%) and four women (22%) and the control group included 14 men (66.7%) and seven women (33.3%). Control subjects had a median of 12 years schooling, whereas the MA group had a median of nine years [Mann–Whitney U = 19.5, *p* < 0.001]. MA abusers reported MA consumption for approximately 15.3 years (SD = 6.1) on average (M_g/week_ = 3.2, SD = 2.3), with a mean age of 17.8 years (SD = 4.0) when beginning the first MA consumption. At the time of measurement, methamphetamine abstinence existed for an average of 8.3 ± 6.4 days. The crystal meth was administrated mostly in an intranasal way and approximately on 16.9 days out of the last 30 days. Both groups reported irregular alcohol consumption with a median of 1×/week. Regarding cannabis consumption, approximately 13.8 years of consumption was reported in the MA group, whereas the control group did not report any cannabis consumption.

All participants received 20 euros for the MRI screening. The local ethics committee of the Friedrich Schiller University, Jena, Germany (# 2019-1545_1-BO) approved the study. To meet ethical requirements, prior to the study, all participants gave their informed consent. Individual data were then saved in accordance with the data protection guidelines (GDRP).

### 2.2. MRI Acquisition

All imaging data were collected on a 3 T whole body system equipped with a 64-element head matrix coil (MAGNETOM PRISMA FIT, Siemens Healthineers, Erlangen, Germany). Firstly, a structural T1 image was acquired followed by a resting-state fMRI to investigate putatively altered functional connectivity (not reported here). Then, the Stroop test was presented in the MR scanner to measure brain activation during cognitive control. For that, a series of 220 whole-brain volume sets were acquired in one session, lasting approximately 8 min. T2*-weighted images were obtained using a multiband multislice GE-EPI sequence (TR = 2120 ms, TE = 36 ms, flip angle = 90°, multiband factor = 4) with 104 contiguous transverse slices of 1.4 mm thickness covering the entire brain and including the lower brainstem. The matrix size was 160 × 160 pixels with an in-plane resolution of 1.4 × 1.4 mm². High-resolution anatomical T1-weighted volume scans (MP-RAGE) were obtained in sagittal orientation (TR = 2.300 ms, TE = 3.03 ms, TI = 900 ms, flip angle = 9°), FOV = 256 × 256 mm², matrix 256 × 256, number of sagittal slices = 192, acceleration factor (PAT = 2) with an isotropic resolution of 1 × 1 × 1 mm³.

### 2.3. The Stroop Task

The manual version of the Stroop test was used in the present study [40], which consisted of two conditions: a congruent (CC) and an incongruent condition (IC). In the CC, 18 color words were presented in the color denoted by the corresponding word; in the IC, 18 color words were displayed in one of three colors, which were not denoted by the word. This target stimulus was presented in the center of the display screen. Two possible answers (color words in black type) were presented to minimize contextual memory demands in the lower visual field. All subjects were instructed to indicate as fast as possible the type of color by pressing one of two buttons (with right index or middle finger), which corresponded spatially to both possible answers. Correct answers were counterbalanced on the right and left sides of the display. Stimulus presentation time was 1500 ms with an interstimulus interval of 10.5 s to allow the hemodynamic response to return to baseline. Additionally, a temporal jitter was introduced to enhance the temporal resolution. Prior to the fMRI measurement, all participants finished 4 practice trials in the scanner. The practice block was repeated until the subjects were able to do them without error.

The Stroop task was implemented using presentation software (Neurobehavioral Systems Inc., Albany, California, http://nbs.neuro-bs.com/ (accessed on 15 May 2019)) running on a PC that was connected to a video projector. The Stroop stimuli were synchronized with the MR scanner and projected onto a transparent screen inside the scanner tunnel, which could be viewed by the subject through a mirror system mounted on top of the MRI head coil. The subjects’ responses were registered by an MRI-compatible fiber optic response device (Lightwave Medical Industries, Richmond, BC, Canada) with two buttons on a keypad for the right hand. This manual version of the Stroop test has been used extensively by our group producing significant and robust activation of the cognitive control network in the brain [40,41,45].

### 2.4. fMRI Data Preprocessing

For the preprocessing of the fMRI data, we used fMRIPrep 20.1.1, which is a preprocessing pipeline, designed to provide an easily accessible, state-of-the-art interface [48]. The functional images were slice-time corrected using 3dTshift from AFNI [49], and corrected for head-motion. There was no significant difference between the groups with respect to the framewise displacement (FD), which is an overall estimate of movement over time (HC: M_FD_ = 0.12; CM: M_FD_ = 0.17). Structural T1 image was adjusted for the intensity non-uniformity and the non-brain tissue was removed. Subsequently, the functional images were then co-registered to the T1 reference using bbregister (FreeSurfer), which implements boundary-based registration [50]. Then, T1 images were normalized to the MNI space by applying nonlinear registration using ANTs (version: 2.2.0). The derived transformation parameters were used to normalize the functional images and afterward spatially smoothed with the Gaussian kernel of 6 mm FWHM (full-width half-maximum) using mcflirt (FSL 5.0.9; [51]). For the statistical analysis, SPM12 software was used (http://www.fil.ion.ucl.ac.uk/spm (accessed on 1 December 2020)). The first four images were discarded. The data were high-pass filtered with a cutoff period of 128 s, corrected for serial correlations choosing AR(1), and analyzed voxel-wise within the general linear model to calculate the statistical parametric maps of t statistics for condition specific effects. As a first step, a fixed-effect model was performed at the single-subject level, which created images of parameter estimates used for the second-level RFX analysis. For that, we set up an ANCOVA design with a between-subjects factor group (MA vs HC), a within-subjects factor task (CC vs. IC), and age as a covariate, and tested for the postulated group differences in the Stroop task. For the whole-brain group comparisons, the statistical comparisons were thresholded on the voxel level at *p* < 0.001 (uncorrected) with a minimum cluster size of k ≥ 16, based on the expected voxels per cluster determined by random field theory in SPM12 software (http://www.fil.ion.ucl.ac.uk/spm, accessed on 1 December 2020) [52].

### 2.5. Behavioral Data Analysis

Behavioral data from the Stroop task were analyzed with SPSS Version 26.0 (https://www.ibm.com/de-de/analytics/spss-statistics-software (accessed on 1 December 2020)). A two-way ANOVA was used with one between-subjects factor group (MA vs. HC), one within-subjects factor condition (CC vs. IC) for the reaction time. Group differences in the number of correct responses were analyzed non-parametrically employing the Mann–Whitney U test.

### 2.6. Correlational Analysis

A Spearman’s rank correlational analysis was performed with the Stoop accuracy and the mean parameter estimates derived from the significant clusters in the IC contrast HC vs. MA in the left DLPFC (x = −37, y = 36, z = 18), anterior midcingulate cortex (aMCC; x = 10, y = 4, z = 32), and V1 (x = 8, y = −73, z = 16). Additionally, we correlated those parameter estimates with the duration of drug consumption, the age of starting crystal meth consumption, and the UPPS total score.

## 3. Results

### 3.1. Behavioral Performance

The two-way ANOVA yielded a significant main effect of condition (F (1,37) = 24.2, *p* < 0.001), indicating a slower response in the IC, compared to CC (Stroop effect). However, no significant main effect of the group or group by condition interaction was detected, the same result was determined when controlling for the effect of age. The Stroop interference time (RT incongruent minus RT congruent condition) for MA abusers (M = 277.0 ms, SD = 415.6) and controls (M = 270.4 ms, SD = 273.6) was similar between groups.

Both groups showed high levels of accuracy in both Stroop task conditions. In total, subjects performed significantly worse in the IC, compared to CC (*p* < 0.001). Healthy controls had 98.7% correct responses in the CC (M_hits_ = 17.8; SD = 0.4) and 91% in the IC (M_hits_ = 16.4; SD = 1.6). MA abusers had 93.2% correct responses (M_hits_ = 16.8; SD = 2.3) in the CC and 83.9% in the IC (M_hits_ = 15.1; SD = 2.8). There was a significant group difference in the total number of correct responses [Mann–Whitney U = 118.0, *p* = 0.041], as well as for the CC [Mann–Whitney U = 121.0, *p* = 0.027], but not for the IC.

Regarding the hypothesized altered impulsivity levels, significant differences were detected between the MA and HC groups in the UPPS subscales: urgency, lack of premeditation, and lack of perseverance (Table 1), and in the total score of the UPPS scale. It showed a higher degree of self-reported impulsivity among the individuals with MA dependence. In contrast, no significant difference was found for the UPPS sensation seeking subscale.

### 3.2. FMRI Data

#### 3.2.1. Group Differences in the Single Stroop Conditions

Individuals with MA dependence demonstrated in the IC a significantly lower fMRI signal bilaterally in the DLPFC (BA9/46), frontal eye fields (FEF, BA8), SMA (BA 6), insula, superior (BA7), and inferior parietal cortices (BA39, BA40), thalamus, in the right aMCC (BA32), as well as in the left caudate and putamen, compared to healthy controls (Figure 1 and Appendix A). In addition, we found significant and large-scale brain activation group differences in several sensory cortical regions, i.e., in the primary (V1) and secondary (V2) visual cortex, fusiform gyrus (BA37), in somatosensory (BA1, BA5) and auditory cortices (BA20, BA21, BA22). As depicted in Table 2, clusters in the occipital cortex, right superior temporal gyrus, left DLPFC, and right insula survived the FWE correction for multiple comparisons at *p* < 0.05.

In the CC, we observed significant, but less pronounced activation differences in the left DLPFC, right VLPFC, bilateral parietal cortex, as well as in several sensory cortical regions (Appendix A). Only one cluster in the occipital cortex survived the FWE correction for multiple comparisons (Appendix A). The MA group did not show any voxels with significantly higher activation, compared to the controls. No significant differences regarding brain activation were found for the group by condition interaction.

#### 3.2.2. Group Differences in the Overall Effect of the Stroop Task

Similar to group differences in the IC contrast, the MA group showed significantly lower activation in the fronto-cingulate and parietal regions and the right thalamus, compared to healthy controls during both Stroop conditions (Appendix A). We also found significant and large-scale brain activation differences in several sensory cortical regions, including the visual cortex, fusiform gyrus, somatosensory, and auditory cortices. Clusters in the occipital cortex, right STG, and VLPFC survived the FWE correction for multiple comparisons (see Appendix A). In contrast, the MA group did not show any voxels with significantly higher activation, compared to the controls.

### 3.3. Correlational Analysis

A significantly positive correlation was detected between the total number of correct responses and the average fMRI signal in the left DLPFC (r = 0.57, *p* < 0.01) and aMCC (r = 0.57, *p* < 0.01) in the MA-dependent individuals, but not in the healthy control group (Figure 2). In an exploratory analysis, the average parameter estimates from the significant cluster in V1/V2 were correlated with the number of correct responses in the Stroop task, which was not significant. No significant correlations were also detected between parameter estimates from the left DLPFC, aMCC, and V1/V2 with the duration of drug consumption, the age of starting crystal meth abuse, and the UPPS total score.

## 4. Discussion

The goal of the present study was to investigate neural correlates of the putatively altered cognitive control processes in individuals with methamphetamine (crystal meth) dependence, using the Stroop task in an event-related fMRI design. We hypothesized a reduced activation in the fronto-cingulo-striatal network in the MA group and associated cognitive control deficits. Our results partly confirm this hypothesis by revealing significantly reduced activation in the dorsal striatum and in the fronto-cingulate cognitive control network, which was significantly correlated with reduced overall behavioral task performance in MA abusers, in terms of lower accuracy. However, the behavioral differences regarding the Stroop effect were not significant between groups. In addition, we observed significantly greater self-reported impulsivity levels in the MA-dependent individuals, compared to healthy controls.

Reduced fronto-cingulate fMRI activation in subjects with MA use disorder is consistent with the previous studies [21,23,26]. This result indicates that chronic crystal meth consumption alters the brain in the frontal, anterior cingulate, and striatal areas, leading to several functional deficits in the domain of cognitive and behavioral control. Our results are also in line with the meta-analysis of Potvin et al. (2018), who reported deficits in individuals with MA use disorder in impulsivity-related functions relative to controls. Moreover, in their systematic review, based on 29 studies, Sabrini et al. (2019) identified altered activation in ACC, PFC, and striatum as the most consistently observed deficits in MA abusers.

Reduced fMRI signal or cerebral blood flow in the fronto-cingulate network during the Stroop task has also been detected in individuals with putatively abnormal NA/DA transmission due to cocaine abuse or in patients suffering schizophrenia, or ADHD [45,53,54,55]. Eckhoff and Wong-Lin [56] showed in nonhuman primates that when LC exhibits high tonic activity and releases high levels of NA, it results in impulsive response and poor accuracy at the behavioral level.

However, the altered brain activation pattern observed in the present study seems to be more generalized and is not exclusively targeting fronto-cingulate brain regions and the cognitive control functions, as revealed by the non-significant behavioral performance in the incongruent Stroop condition. The pharmacologic action of crystal meth may explain this observation since both noradrenaline and dopamine are crucially involved in modulating several brain states subserving attentional, reward-related, cognitive control, and memory processes [57,58,59,60] by selectively optimizing task-relevant behavioral responses [61]. Furthermore, given the widespread neural projection of the NA and DA producing nuclei in the brainstem and midbrain, it is conceivable that chronic abuse of MA might affect various cognitive domains, not only those related to the cognitive control functions.

Along with widespread alterations at the neural level, we observed a significantly decreased accuracy in the total number of correct responses in the MA abusers, which was significantly related to decreased fMRI activation in the DLPFC and aMCC. This result confers our study hypothesis and demonstrates the impact of MA on fronto-cingulate brain activation during the Stroop task and associated poor overall task performance.

However, no differences in reaction times were detected in both Stoop conditions and in the Stroop interference time between groups, indicating that the MA-dependent individuals were not slowed in the present study, even regarding the Stoop effect. This finding contradicts some of the previous studies, reporting prolonged reaction times, not only in the Stroop task [20] but also in other neuropsychological tests on cognitive control functions, such as decision-making [19] and motor inhibition assessed with the stop signal task [26]. In another study, abstinent people with MA use disorder displayed significant differences in the Stroop interference time, even compared to long-term abstinent ones [43]. However, Jan et al. (2014) and Farhadian and Akbarfahimi [35] also found no significant difference in the interference scores between MA-dependent individuals and controls, which is in line to our findings.

A potential explanation might be a lack of coordinated activation in the task-related neural networks as a consequence of a disruption in synchronized activity due to MA consumption [62]. A supposed function of dopamine, e.g., in the ACC or PFC, is modulation of high-frequency neural synchronization and thus optimizing information processing during cognitive tasks [63]. This might lead to more incorrect responses on the behavioral level, as observed in the present investigation, but not necessarily to prolonged reaction times. Furthermore, differences in patient’s characteristics, such as the clinical or cognitive status and duration of MA consumption may explain the deviating results regarding reaction times. The clinical status of the subjects varied from study to study, including individuals in early abstinent [33], late abstinent [20], active users [36], or adolescent MA abusers [32].

Surprisingly, as a new finding, we demonstrated significantly lower and pronounced brain activation in the MA group in several sensory cortical regions, i.e., in the visual, auditory and somatosensory cortices during the Stroop task. It appears that brain networks processing sensory information from different modalities are significantly affected by chronic crystal meth consumption.

Previously, a single dose of methamphetamine was shown to hyperactivate the auditory cortex during a tone discrimination task and primary sensorimotor cortex during a finger-tapping task as a supposed primary effect of the drug [64]. Another fMRI study investigating neural responses to visual stimuli revealed that when presenting MA-paired stimuli in the scanner, it produced greater activation in regions related to visual and auditory processing, compared to placebo-paired stimuli [65]. Thus, individuals regularly taking MA may permanently hyperactivate those brain regions, showing consequently blunted responses to visual or auditory stimuli. Another study tested the effects of methylphenidate (MPH) on brain activation during the Stroop task in active methamphetamine dependence and observed higher fMRI activation in the superior occipital gyrus, superior parietal gyrus, middle occipital gyrus, and inferior parietal lobule, when comparing the MPH group to the placebo group after the drug administration in MA abusers [36].

Furthermore, it has been shown, that both DA and NA systems, are involved in remodeling the tuning properties of the sensory neurons, modulating their intrinsic currents and hence excitability [66]. In addition to the complex pharmacological mechanisms, the effects of MA on dopamine neuron excitability and output have been shown to be concentration dependent. For instance, MA increased dopamine neuron firing at low doses and enhanced stimulated dopamine neurotransmission, whereas at higher concentrations, both effects were reversed [10]. Furthermore, a recent study demonstrated reduced type II pyramidal cell excitability in the medial PFC after multiple MA administrations [11]. Thus, we can speculate, that the putative effect of chronic MA consumption on sensory brain regions could be in changing the firing properties of those neurons and leading to the observed blunted activation in sensory brain regions, notably in the visual cortex, due to visual stimuli presentation in the present study. Future research should specifically examine changes in the excitability of neurons in sensory brain regions, as well as putative changes in sensory processing in individuals with chronic MA use.

Thus, although we did not observe any direct relationship between the Stroop task performance and fMRI activation in sensory brain regions, it is conceivable that altered visual, and not exclusively executive functioning in MA abusers, might be related to the observed reduced overall cognitive performance, which was not IC specific. Once visual processing is altered during the Stroop task, it might cause changes in all successive higher-order cognitive processes.

In addition, because fMRI activation parameters are computed relative to an arbitrary baseline, the reduced activation in sensory cortices might be caused by a high sustained baseline activity due to the chronic MA consumption. This might limit the further increase in, e.g., visual cortex activation (ceiling effect) during the processing of the Stroop stimuli. This may explain our findings of a significantly reduced BOLD signal in visual and auditory cortices in MA abuser subjects, however, it would be deserving to further examine the effects of methamphetamine abuse on sensory information processing in more detail since it has not been studied until now.

Finally, some limitations of the study must be mentioned. Firstly, the relatively small sample size of the study should be mentioned. The statistical analyses were reported with the conservative, but uncorrected, *p*-value of 0.001. However, several clusters in the left DLPFC, VLPFC, temporal, and occipital regions remained significant after the FWE correction. Secondly, the groups differed in education level. The control group had a higher school education than the MA group. The level of education could influence the speed-dependent Stroop test score, as reported in the previous study [67], however, we did not find any differences in the reaction times in the Stroop task between groups. We therefore consider the effect of education on performance in the Stroop task to be negligible. Furthermore, individuals with methamphetamine use disorder reported cannabis consumption, but not dependence, which cannot be ruled out as a potential confounder. We did not collect any precise data of cannabis consumption for the controls, but they were screened for the current use of alcohol and drugs. We also did not perform toxicology screenings to examine the sobriety of the healthy control group, but we checked for sobriety using a clinical interview. Furthermore, subjects who fulfilled the criteria of cannabis abuse or dependence according to DSM-IV, were excluded from the study.

## 5. Conclusions

In this event-related fMRI study, we observed decreased brain activation during the Stroop task performance in the fronto-cingulate, parietal, and striatal regions, but also, as a new finding, in several sensory cortical regions in MA abusers relative to healthy controls. Together with significant correlations between BOLD signals in aMCC and DLPFC and the overall impaired task performance, these results provide further evidence for the neural basis of the frequently reported altered cognitive function in MA users, in terms of cognitive control and decision making. As a new finding, we also revealed abnormal activation in several sensory brain regions, suggesting the negative effect of MA use on the proper neural activity of these regions. Future research should specifically examine the neurotoxic effects of MA on sensory processing in individuals with chronic MA use.

## Figures and Tables

**Figure 1 brainsci-13-00197-f001:**
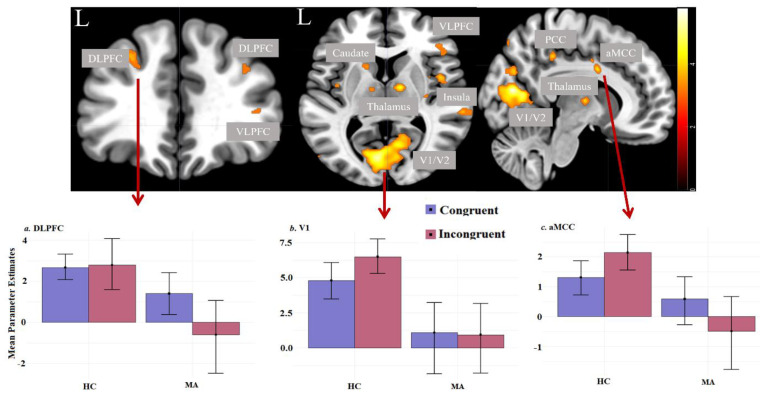
Significant group differences (healthy controls vs. individuals with MA dependence) in brain activation during the incongruent Stroop task condition (voxel-level: *p* < 0.001 uncorr., cluster-level: corrected according to expected voxels per cluster ≥ 16). The bar graphs depict parameter estimates extracted from the significant cluster in the left DLPFC, aMCC, and V1. (**a**) Averaged parameter estimates, and standard error extracted from the significant cluster in the left DLPFC (local maximum: x = −37, y = 36, z = 18, cluster size = 65); (**b**) averaged parameter estimates and standard error extracted from the significant cluster in the V1 (local maximum: x = 8, y = −73, z = 16, cluster size = 3482); (**c**) averaged parameter estimates and the standard error extracted from the significant cluster in the aMCC (local maximum: x = 10, y = 4, z = 32, cluster size = 45). Abbreviations: HC, healthy controls; MA, individuals with methamphetamine dependence; DLPFC, dorsolateral prefrontal cortex; VLPFC, ventrolateral prefrontal cortex; aMCC, anterior midcingulate cortex; V1, primary visual cortex; V2, secondary visual cortex; SMA, supplementary motor area.

**Figure 2 brainsci-13-00197-f002:**
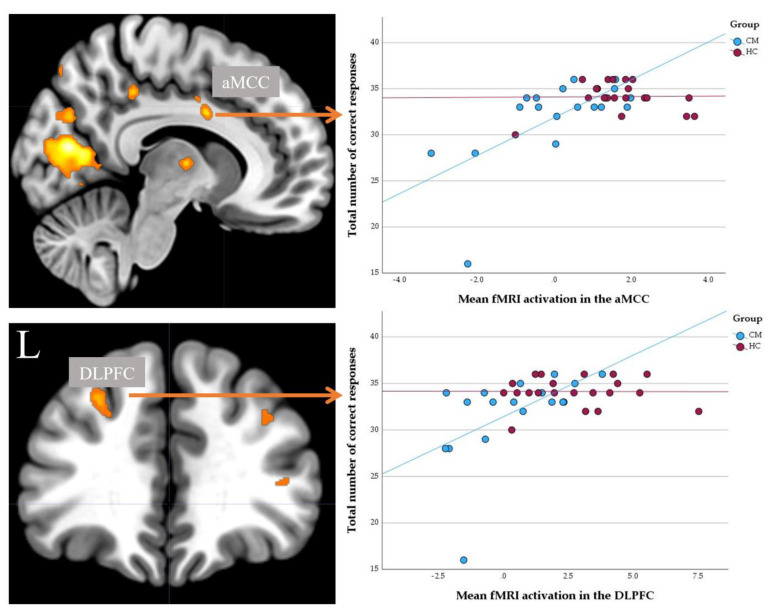
The scatterplot depicts significantly positive correlations between the fMRI signal in the aMCC (individuals with MA dependence: r = 0.57, *p* < 0.01), left DLPFC (individuals with MA dependence: r = 0.57, *p* < 0.01), and the total number of correct responses in individuals with methamphetamine dependence (MA; blue color) and healthy controls (HC; red color).

**Table 1 brainsci-13-00197-t001:** Differences between individuals with methamphetamine dependence and healthy controls, regarding self-reported impulsivity, as assessed with the UPPS scale.

Subscale	Group	Mean	SD	*t*	df	*p*-Value
Urgency	MA HC	30.8 23.6	7.57 5.99	3.298	36	0.002 *
Lack of Premeditation	MA HC	25.4 21.9	4.02 3.50	2.916	36	0.006 *
Lack of Perseverance	MA HC	20.7 17.0	4.55 2.89	3.013	36	0.005 *
Sensation Seeking	MA HC	35.5 37.9	8.07 7.64	−0.933	36	0.357
Total Score	MA HC	112.4 100.3	11.61 11.46	3.211	36	0.003 *

* *p*-values survive the Bonferroni correction.

**Table 2 brainsci-13-00197-t002:** Maxima of regions showing significantly reduced fMRI signals in individuals with crystal meth dependence, compared to healthy controls in the incongruent Stroop task condition (voxel-level *p* < 0.001 uncorr., cluster size ≥16, according to the expected voxels per cluster).

					MNI Coordinates			
Region of Activation	Left/Right	Brodmann Area	Cluster Size	P_FWE-corr_ (Cluster-Level)	x	y	z	t Value
Occipital Cortex	R	17	3482	0.000	8	−73	16	5.82
Occipital Cortex	R	19	163	0.047	16	−61	−7	5.03
Superior Temporal Gyrus	R	22	296	0.003	54	−25	2	4.94
DLPFC	L	9	269	0.005	−27	40	38	4.74
Insula/VLPFC	R	13/44	174	0.037	44	4	6	4.06
Premotor Cortex	R	6	120	0.132	32	−13	54	4.70
Parahippocampal Gyrus	R	36	23	0.962	16	−37	−15	4.53
Occipital Cortex	L	19	305	0.002	−43	−65	2	4.49
Frontal Eye Field	R	8	56	0.602	18	30	56	4.48
Thalamus	R		42	0.779	12	−7	6	4.48
Motor Cortex	R	4	41	0.792	60	−13	46	4.45
Medial Temporal Gyrus	L	21	58	0.578	−69	−31	−13	4.41
Fusiform Gyrus	R	37	105	0.190	42	−35	−19	4.36
Premotor Cortex	L	6	63	0.519	−23	15	52	4.35
Supramarginal Gyrus	L	40	31	0.902	−45	−33	26	4.30
Somatosensory Cortex	L	1	99	0.220	−59	−15	32	4.29
Supramarginal Gyrus	L	40	97	0.232	−49	−39	56	4.27
aMCC	R	24	45	0.742	10	4	32	4.26
Premotor Cortex	L	6	22	0.968	−59	12	40	4.15
Superior Temporal Gyrus	L	22	32	0.892	−47	−41	14	4.11
Cerebellum	L		137	0.087	−25	−35	−47	4.10
Premotor Cortex	L	6	20	0.977	−35	−17	60	4.09
Dorsal PCC	R	31	23	0.962	10	−37	42	4.08
DLPFC	R	9	105	0.190	36	50	30	4.07
Angular Gyrus	L	39	92	0.262	−33	−49	34	4.06
Hypothalamus	R		30	0.911	2	−3	−13	4.05
DLPFC	R	9	52	0.652	38	32	30	4.03
Fusiform Gyrus	L	37	29	0.920	−17	−35	−25	3.96
Insula	L	13	37	0.839	−33	−9	12	3.96
Frontal Eye Field	R	8	44	0.754	40	32	46	3.95
Superior Parietal Lobule	L	7	44	0.754	−33	−61	46	3.95
Inferior Temporal Gyrus	L	20	48	0.703	−47	−9	−33	3.94
Putamen	L		33	0.882	−19	−5	−6	3.91
Premotor Cortex	R	6	23	0.962	44	−3	48	3.90
Insula	R	13	30	0.911	30	−15	14	3.88
Cerebellum	L		16	0.990	−15	−45	−53	3.88
DLPFC	L	46	65	0.496	−37	36	18	3.87
Premotor Area	L	6	36	0.850	−31	10	64	3.87
Angular Gyrus	R	39	45	0.742	62	−45	28	3.86
RLPFC	L	10	30	0.911	−25	56	18	3.76
Superior Parietal Lobule	L	5	40	0.804	−29	−43	56	3.76
Premotor Cortex	L	6	40	0.804	−5	−19	70	3.76
Fusiform Gyrus	L	37	21	0.972	−25	−29	−31	3.75
Caudate	L		28	0.928	−17	14	8	3.75
Frontal Eye Field	R	8	37	0.840	2	14	38	3.75
Motor Cortex	L	4	36	0.850	−59	−7	38	3.71
Cerebellum	R		90	0.275	24	−47	−51	3.71
Supramarginal Gyrus	R	40	16	0.990	68	−15	28	3.70
Thalamus	L		33	0.882	−15	−15	12	3.68
Superior Parietal Lobule	R	7	72	0.423	16	−75	50	3.64
Superior Parietal Lobule	R	7	37	0.839	16	−65	50	3.60
Supramarginal Gyrus	R	40	32	0.892	56	−29	48	3.60
DLPFC	R	46	19	0.981	44	34	8	3.52
Superior Parietal Lobule	L	5	18	0.984	−13	−31	48	3.50

Abbreviations: R, right; L, left; DLPFC, dorsolateral prefrontal cortex; RLPFC, rostrolateral prefrontal cortex; PCC, posterior cingulate cortex; aMCC, anterior midcingulate cortex; PCC, posterior cingulate cortex.

## Data Availability

The original contributions presented in the study are included in the article, further inquiries can be directed to the corresponding author/s.

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
