# Peer review of "Neural Correlates of Impaired Cognitive Control in Individuals with Methamphetamine Dependence: An fMRI Study"

_brainsci, 2023, doi:10.3390/brainsci13020197_

Round 1

Reviewer 1 Report

In the current manuscript, Zerekidze et al investigated the neural correlates of impaired cognition in individuals with methamphetamine (MA) dependence using Stroop task, fMRI and an impulsivity questionnaire. The authors reported poor accuracy on the Stroop task and higher self-rated impulsivity. fMRI data revealed decreased activation in dlPFC, aMCC and dorsal striatum in MA dependent individuals versus healthy controls. Alterations in fMRI signal in dlPFC and aMCC significantly correlated with impaired behavioral task performance in individuals with MA dependence. Significantly lower and pronounced activation was detected in sensory cortices. The authors claimed that abnormal activation in sensory cortices due to MA chronic use could be attributed to negative effect of its use on neural activity in these regions. The manuscript is timely and adds novelty on the negative impact of meth on sensory processing brain regions. This work will add to the expanding literature of MA dependence on sensory processing.

I have some questions and suggestions for the authors, mostly hovering around the biological significance of the current work. I have a couple of suggestions that the authors could include/clarify in the introduction:

1.      The authors should clearly explain how MA blocks presynaptic reuptake of monoaminergic NTs. A couple of lines on this will be helpful.

2.      The explanation given between lines 48-51 about increased NA transmission could be framed better. The authors need to state clearly whether LC neurons projecting to ACC release more NA or if there is dense projection from LC to ACC that attributes to increased NA release.

3.      The authors claim that DA and NA systems are involved in remodeling the tuning properties of sensory neurons, modulating their intrinsic properties and excitability – is it known if chronic MA use blunts intrinsic excitability of neurons in sensory cortices? If yes, how are DA and NA systems modulated by meth so that there is a direct implication of intrinsic excitability change?

There are grammatical and spelling errors in the manuscript which should be corrected. 

Author Response

We would like to thank both reviewers for their helpful comments which have led to a substantial improvement of our manuscript. As described below, we adopted all suggestions.

Reviewer #1

The authors should clearly explain how MA blocks presynaptic reuptake of monoaminergic NTs. A couple of lines on this will be helpful.

Reply: We thank the reviewer #1 for this helpful suggestion. We added the following paragraph in the introduction section on pages 1-2.

“Previous studies showed that MA consumption leads to a marked increase in the levels of monoaminergic neurotransmitters: the dopamine (DA), noradrenaline (NA), and serotonergic (5-HT) in the CNS, as well as in the peripheral nervous system by multiple complex pharmacological mechanisms [1, 2]. It inhibits monoamine reuptake transporters as well as reverses transport of neurotransmitter through plasma membrane transporters [3]. MA also inhibits the monoamine oxidase activity and increase the activity and expression of the tyrosine hydroxylase, which catalyzes the conversion of the amino acid L-tyrosine into levodopa, a DA precurser [4]. These mechanisms lead in total to a significant release of monoamines.”

The explanation given between lines 48-51 about increased NA transmission could be framed better. The authors need to state clearly whether LC neurons projecting to ACC release more NA or if there is dense projection from LC to ACC that attributes to increased NA release.

Reply: We rephrased these sentences accordingly and stated on page 2:

“Furthermore, high density of noradrenergic (NA) fibers from NA-synthesizing neurons in locus coeruleus (LC) to the anterior cingulate cortex (ACC), the prefrontal cortex (PFC) and the hippocampus have been shown [5-8]. Thus, a massive release of NA after MA consumption would strongly affect the excitability of these regions, having an effect on arousal, memory, attention, and cognitive control processes [6-9].”

The authors claim that DA and NA systems are involved in remodeling the tuning properties of sensory neurons, modulating their intrinsic properties and excitability – is it known if chronic MA use blunts intrinsic excitability of neurons in sensory cortices? If yes, how are DA and NA systems modulated by meth so that there is a direct implication of intrinsic excitability change?

Reply: We thank reviewer #1 for this comment.

Indeed, we have limited understanding of how MA alters neural responses to sensory stimuli in chronic abuse. As already stated in the manuscript, there are some few studies which reported increased brain activation in sensory cortices after MA administration. Beside the complex pharmacological mechanisms, the effects of MA on dopamine neuron excitability and output have been additionally shown to be concentration dependent. For instance, MA increased dopamine neuron firing at low doses and enhanced stimulated dopamine neurotransmission, whereas at higher concentrations, both effects were reversed [10]. Furthermore, a recent study demonstrated reduced type II pyramidal cell excitability in the medial PFC after multiple MA administration [11]. Thus, we can speculate, that the putative effect of MA on senso-ry brain regions could be in changing the firing properties of that neurons and leading to the observed blunted activation in sensory brain regions, notably in the visual cortex due to visual stimuli presentation in the present study. Future research should specifically examine changes in the excitability of neurons in sensory brain regions as well as putative changes in sensory processing in individuals with chronic MA use. We modified the discussion section accordingly.

There are grammatical and spelling errors in the manuscript which should be corrected. 

Reply: Thanks for the suggestion. The paper went through the thorough grammar and spelling check.

Reviewer 2 Report

Thank you for inviting me to review this manuscript.
In this study the authors aim to explore the neural correlates of impaired cognitive control in individuals with methamphetamine (MA) dependence according to DSM-IV criteria. The results of the current study provide evidence for the negative impact of chronic crystal meth consumption on the proper functioning of the fronto-cingulate and striatal brain regions, presumably underlying the often-observed deficits in executive functions in individuals with MA use disorder. 

The study is potentially interesting, but some points remain to be clarified: 

1) T-value for sensation seeking in table 1 looks wrong.
2) I don't understand how it was ruled out that the group of MA users did not use other drugs in the past and for how long.
3) No PFWE-corr result found? from Table 1 of the Supplementary Materials it would appear that Occipital Cortex, Superior Temporal Gyrus, DLPFC and Insula/VLPFC could survive. If not, you could try ROI analysis in some regions. However, I would suggest inserting a summary table in the paper because not everything was clear to me.
4) There are no conclusions explaining the importance of the results obtained and any future prospects.

Author Response

We would like to thank both reviewers for their helpful comments which have led to a substantial improvement of our manuscript. As described below, we adopted all suggestions.

Reviewer #2

T-value for sensation seeking in table 1 looks wrong.

Reply:  We thanks reviewer #2 for the careful reading of the manuscript. Indeed, there was a slight numerical error in the decimal point range. We corrected the t-value for sensation seeking in the revised table 1.

I don't understand how it was ruled out that the group of MA users did not use other drugs in the past and for how long.

Reply: We stated no on page #4 of the revised version of the manuscript.

“Furthermore, patients, who fulfilled the criteria of other substance dependence in the last 12 months, were excluded from the study.”

No PFWE-corr result found? from Table 1 of the Supplementary Materials it would appear that Occipital Cortex, Superior Temporal Gyrus, DLPFC and Insula/VLPFC could survive. If not, you could try ROI analysis in some regions. However, I would suggest inserting a summary table in the paper because not everything was clear to me.

Reply: We thank the reviewer #2 for this very important comment. Now we inserted the table 2 into the revised version of the manuscript, indicating the clusters which will survive the FWE correction. We also updated and corrected the supplementary tables regarding this information. We apologize that in the first version of the manuscript the presentation of the FWE corrected values was misleading.

There are no conclusions explaining the importance of the results obtained and any future prospects.

Reply: We added the “conclusion” section to the revised version of the manuscript.

“In this event-related fMRI study, we observed decreased brain activation during Stroop task performance in the fronto-cingulate, parietal and striatal regions, but also, as a new finding in several sensory cortical regions in MA abusers relative to healthy controls. Together with significant correlations between BOLD signals in aMCC and DLPFC and overall impaired task performance, these results provide further evidence for the neural basis of the frequently reported altered cognitive function in MA users in terms of cognitive control and decision making. As a new finding, we also revealed abnormal activation in several sensory brain regions, suggesting the negative effect of MA use on the proper neural activity of these regions. Future research should specifically examine the neurotoxic effects of MA on sensory processing in individuals with chronic MA use.”

Round 2

Reviewer 2 Report

The paper can be accepted in this form.